

# Genome-wide sequence identification and expression analysis of ARF family in sugar beet (*Beta vulgaris* L.) under salinity stresses

Jie Cui, Xinyan Li, Junliang Li, Congyu Wang, Dayou Cheng and Cuihong Dai

Harbin Institute of Technology, Harbin, China

## ABSTRACT

Auxin response factor (ARF) proteins respond to biological and abiotic stresses and play important roles in regulating plant growth and development. In this study, based on the genome-wide database of sugar beet, 16 BvARF proteins were identified. A detailed investigation into the BvARF family is performed, including analysis of the conserved domains, chromosomal locations, phylogeny, exon-intron structure, conserved motifs, subcellular localization, gene ontology (GO) annotations and expression profiles of BvARF under salt-tolerant condition. The majority of BvARF proteins contain B3 domain, AUX_RESP domain and AUX/IAA domain and a few lacked of AUX/IAA domain. Phylogenetic analysis suggests that the 16 BvARF proteins are clustered into six groups. Expression profile analysis shows that most of these BvARF genes in sugar beet under salinity stress were up-regulated or down-regulated to varying degrees and nine of the BvARF genes changed significantly. They were thought to have a significant response to salinity stress. The current study provides basic information for the BvARF genes and will pave the way for further studies on the roles of BvARF genes in regulating sugar beet's growth, development and responses to salinity stress.

## INTRODUCTION

ARF are transcription factors that activate or inhibit the expression of auxin-response genes by binding to specific positions in the promoter region of auxin-responsive genes, thereby regulating plant growth and development, including vascular elongation, cell division, apical dominance, flowering, fruit development (*Chandler, 2016*; *Fleming, 2006*; *Kumar, Tyagi & Sharma, 2011*; *Li et al., 2016*; *Ljung, 2013*; *Su et al., 2014*), and responding to abiotic stresses. Most of the ARF proteins consist three conserved domains: the N-terminus DNA binding domain (DBD), the middle region (MR), and the C-terminus domain (PB1). Crystal structures of the DBDs of ARF1 and ARF5 highlight the presence of three different subdomains: a B3 subdomain involved in the recognition of the ARF-specific AuxRE DNA motif, a dimerization domain (DD) allowing ARF dimerization, and a Tudor-like ancillary domain (AD) of unknown function which might be involved in an interaction

Corresponding author
Jie Cui, cuijie@hit.edu.cn

with the DD (*Roosjen, Paque & Weijers, 2018*). The B3 DNA binding domain is capable of binding to a TGTCTC/GAGACA site in the promoter region of the target genes. And ARF proteins homodimerize through their DD, which were also called AUX_RESP domain in ARF, by hydrophobic interactions (*Boer et al., 2014*). The MR domain has an activation domain (AD) or a repression domain (RD) activity, thereby having an effect of activating or inhibiting transcription. The PB1 domain is the region of homologous dimerization and heterologous dimerization of ARFs or AUX/IAA proteins and ARF (*Yu et al., 2014*).

Since the first *Arabidopsis* ARF gene, ARF1, was cloned and its function investigated (*Ulmasov, Hagen & Guilfoyle, 1997*), the research on ARF of *Arabidopsis* has entered into the molecular mechanism of regulating plant growth. In recent years, Krogan and others had identified the direct regulation relationship between genes that play a major role in auxin signaling and trafficking (*Krogan et al., 2016*). Studies by *Liu et al. (2014)* showed that ARF 3 promotes flower meristem formation by inhibiting WUS expression. Zhao demonstrated that the transcription factor ARF2 modulates the expression of the $K^+$ transporter gene HAK5 in *Arabidopsis thaliana*, and found that AtARF2 responds to low potassium stress (*Zhao et al., 2016*). *Zhang et al. ( 2014)* studied in AtARF5 and found that it could up-regulate STOMAGEN significantly to regulate stomatal development. *Li, Dai & Zhao, (2006)* showed that the transcription factors ARF7 and ARF19 are not only important for auxin signaling, but also play critical roles for *Arabidopsis* to respond to ethylene. It may indicate that AtARF7 and AtARF19 may be involved in the interaction of the ethylene pathway and the auxin pathway. With the deepening of research on model plant ARF, the research on ARF of other plants has also started. There are articles on the whole gene analysis of plant ARF such as maize (*Xing et al., 2011*), sweet orange (*Li et al., 2015*), rice (*Wang et al., 2007*), banana (*Hu et al., 2015b*) and grape (*Wan et al., 2014*). In these studies, ARF were divided into six classes according to their homology to *Arabidopsis thaliana* ARF. Some members of these six classes are able to respond to one or more abiotic stresses. It is worth noting that in the study of physic nut ARF (*Tang et al., 2018*), seven ARFs such as JcARF2, JcARF11, JcARF14 and JcARF15 showed significant changes in expression under salt stress conditions. These JcARFs belong to different five classes, so it is speculated that ARF may be generally involved in the response of salt stress. A study in tea trees (*Xu et al., 2016*) has shown that ARF members in roots and shoots of tea plants respond to salt stress, and CsARF6 shows significant up-regulation under different stress conditions, while CsARF2-1 and CsARF11 show significant down-regulation. It is speculated that CsARF members may participate in the regulation of salt tolerance through different ways.

Sugar beet is a biennial herb of *Chenopodiaceae*. It is an important sugar-producing crop and widely used in food, sugar production, feed, etc. The yield and quality of sugar beet are susceptible to various biologic and abiotic stresses, especially the soil salinity. In China, the sugar beet production area and the saline land overlap in a large area. Even though sugar beet is a salt-tolerant plant, it is limited in extent. The high salinity of the land not only decreases the suitable planting area of sugar beet but also reduces the yield per unit area. In general, the study of salt-tolerant transcription factors contributes to the cultivation of salt-tolerant related sugar beets and lays the foundation for increasing total sugar beet yield. ARF genes participate in responses to abiotic stresses (*Qi et al., 2012*; *Shen et al.,*

*2013*), however, there is no genome-wide analysis for BvARF. Dohm et al. (*2012*; *2014*) constructed the whole genome of sugar beet. The physical map and sequence of the whole genome of sugar beet make it possible to study the salt tolerance of sugar beet from the genetic level. Based on the whole genome database of sugar beet, the genome-wide analysis of BvARF was carried out by bioinformatics method, and the ARF related to salt treatment was identified, which provide a basis of breeding salt-tolerant sugar beet varieties.

## MATERIALS AND METHODS

### Materials

The research materials used in this experiment for salinity stress were "O68" strain, which was the salt-tolerant sugar beet strain selected by the laboratory (*Shi et al., 2008*). In the sugar beet seedling stage, the sugar beet seeds were first soaked in running water for 12 h and then sown into the wet sponge which was placed in the incubator for 2 days (dark, 24 °C). Sugar beet seedlings were taken out after they germinated and placed in the culture pot at $22 \pm 1$ °C with 16 h/d supplemental lighting, followed by 8h/d darkness, and cultured with clear water until they grow a third pair of true leaves, then they were treated with salinity stress. The salt stress conditions were that the nutrient solution was replaced with a 300 mM NaCl solution for 24 h, the remaining conditions were unchanged, and the control plants were set at the same time without salt treatment. After the stress treatment, leaves and roots were collected. And they were wrapped in tin foil, quickly placed in liquid nitrogen for precooling. The frozen samples were uniformly stored in the ultra-low temperature refrigerator ($-80$ °C).

### BvARF identification in sugar beet

The whole genome data of sugar beet was published by Max Planck Institute for Molecular Genetics in 2014 (http://bvseq.molgen.mpg.de/index.shtml). And the assembly RefBeet-1.1 of KWS2320 was downloaded for bioinformation research. It includes functional annotation, genome sequence, transcript sequences, and protein sequences etc. ARF proteins conserved domain (PF06507) on Pfam were used as a seed sequence to search the whole genome data, and $e$-value $< 1e^{-5}$ was set on HMMER (http://www.hmmer.org/). Pfam online tool performed a conserved domain analysis of candidate BvARF protein sequences (*Finn et al., 2016*), screening protein sequences containing the B3 domain (PF02362), AUX_RESP domain (PF06507) and AUX/IAA (PF02309) domains to remove redundant proteins, and remainders were considered to be BvARF proteins. DNAMAN7.0 was used for multiple sequences alignment of the full-length coding sequence of BvARF proteins, and the conserved domains in the BvARF protein sequences were identified by Weblogo2.8.2 (http://weblogo.berkeley.edu/logo.cgi).

### Bioinformatics analysis of BvARF family

Molecular weight and isoelectric point of BvARF proteins were predicted by ExPASY (https://web.expasy.org/protparam/). MapInspect was used to map the position of the BvARF genes on chromosomes. Exon and intron structures of the BvARF genes were analyzed by GSDS (http://gsds.cbi.pku.edu.cn/Gsds_about.php) (*Hu et al., 2015a*).

**Table 1  Primer sequences of ARF.**

| Gene | Forward primer(5′–3′) | Reverse primer(5′–3′) |
| --- | --- | --- |
| ghtj | CTGTGTCCACTGACCTAAA | ATCTCTGAGCACTAAGCCC |
| qzmp | CAAGATTTCTGTAGTCCCG | GTCTCCAGTATTTTGTCCC |
| hgze | GTGTGGCGATAAGCAGAATAG | TCCTTGCCTTTGTTTCCTGTA |
| qfwi | GCTCAGATGACACTCCTACC | AACAAATAGACTCCATCCTG |
| jrpi | TGTTCGTTGGGATTCGGAGG | ATTCCGCTCGCACTTTCTCA |
| okdq | AAGCCTTGTTATCATCCG | GGTTTGCTAGTCCCTCGT |
| yzaj | ATTGAAGGCTGAAGCGGATAC | CGTCTGAGCACCGAGAACC |
| orwr | ATTCAGGTGGAGTTGATGTT | AGGCTTTAGTGGTTCAGTTT |
| kddw | ACGCCACCTACTGACTACA | CCTCATACCAACAGAAAGC |
| eqms | CCGAGTTTGTTGTGAAGGC | GAAGGGGGACAGATGAATG |
| opag | TTCGAGGGTGTCCAAGTTCC | GCGGCAGGAACGGTAGAATA |
| zzhs | AGGGCAACCAAAACGACACT | TCCTAAATCGCATCCCAAGA |
| gcik | GCTTGTATTTGTTGACCG | ACTTCTTGAGGGGATAGG |
| efdx | CGGTATTGTTGTTGGTGTTA | ACATCCATAGGGAGGTGA |
| ICDH | CACACCAGATGAAGGCCGT | CCCTGAAGACCGTGCCAT |

ClustalX was used to carry out a multiple sequence alignment of BvARF proteins (*Larkin et al., 2007*). A phylogenetic tree of BvARF family and AtARF family and a phylogenetic tree of BvARF family were constructed by the neighbor-joining with bootstrap replicate set to 1000, and the other parameters default. And the protein sequences of AtARF for comparison were provided in File S1. BvARF proteins were submitted to MEME (http://meme-suite.org/tools/meme) in the order of homology and subjected to motif analysis (*Bailey et al., 2006*), in which the number of expected motifs was set to 20, and the rest parameters were all default values. Subcellular localization of BvARF proteins was predicted by using CELLO (http://cello.life.nctu.edu.tw/). Gene Ontology (GO) categories of genes were obtained from PlantTFDB (http://planttfdb.cbi.pku.edu.cn/), and GO level 2 of BvARF genes were visualized with WEGO (http://wego.genomics.org.cn/). And the GO annotation numbers of BvARF were provided in File S2.

## Expression analysis of BvARF under salinity stress

Total RNA was isolated from the aforementioned sugar beet samples using MiniBEST Plant RNA Extraction Kit (TaKaRa, Japan) and reverse transcribed into cDNA using High Capacity cDNA Reverse Transcription Kit (applied biosystems, USA). The patterns of expression of these genes under normal growth conditions and in response to salinity stresses were analyzed by quantitative real-time PCR (qRT-PCR). Primers for qPCR were designed using Primer 5 and NCBI Primer-BLAST, synthesized by Huada Gene Co., Ltd., and the primer sequences are shown in Table 1. The ICDH (Isocitrate dehydrogenase) gene was used as an internal control.

For quantitative analysis using the CFX96 Real-Time System (BIO-RAD, USA), and iTaq Universal SYBR Green Supermix kit (BIO-RAD, USA) was used with ICDH as the housekeeping gene. The reaction system was 10 μL, in which the dye was 5 μL, the ddH2O was 3.4 μl, the forward primer and reverse primers were each 0.4 μL, and the cDNA

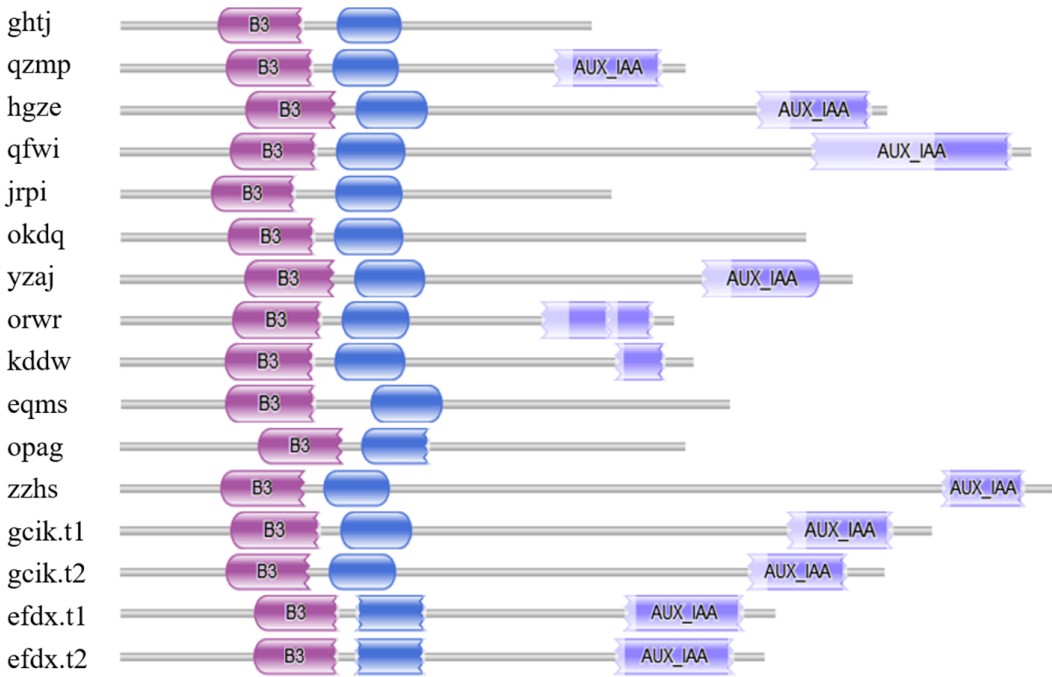

**Figure 1 Conservative domains analysis of BvARF.** The purple color block on the left represents the B3 domain, the blue color block in the middle represents the AUX_RESP domain, and the medium slate blue color block on the right represents the AUX/IAA domain.

was 0.8 µL, and each treatment was repeated three times. Data analysis was performed by calculating $2^{-\Delta\Delta T}$, and the relative expression amount of each gene was expressed by mean ± standard deviation.

## RESULTS

### Identification of BvARF family members

Seventeen candidate BvARF proteins were obtained by using PF06507 as a seed sequence to perform HMMER alignment in the sugar beet genome database. And the sequences of 17 candidate BvARF proteins were analyzed by Pfam for conserved domain, and the proteins containing no specific domains of ARF were removed. A total of 16 BvARF proteins were identified, as shown in Fig. 1.

It can be seen from the figure that all BvARF have a C-terminal B3 binding domain and an intermediate AUX_RESP domain, while the AUX/IAA domain exists only in 11 BvARF except for opag, okdq, ghtj, jrpi, and eqms.

From the comparison results of DNAMAN7.0, the B3 DNA binding domain is highly conserved and occasionally modified. The AUX_RESR domain is not as conserved as B3, the AUX/IAA domain is the least conserved and in short length, as shown in Fig. 2.

In the 16 BvARF proteins, gcik.ti and gcik.t2 were both translation products of transcript variants of BvARF gene gcik. And the efdx.t1 and efdx.t2 were both translation products

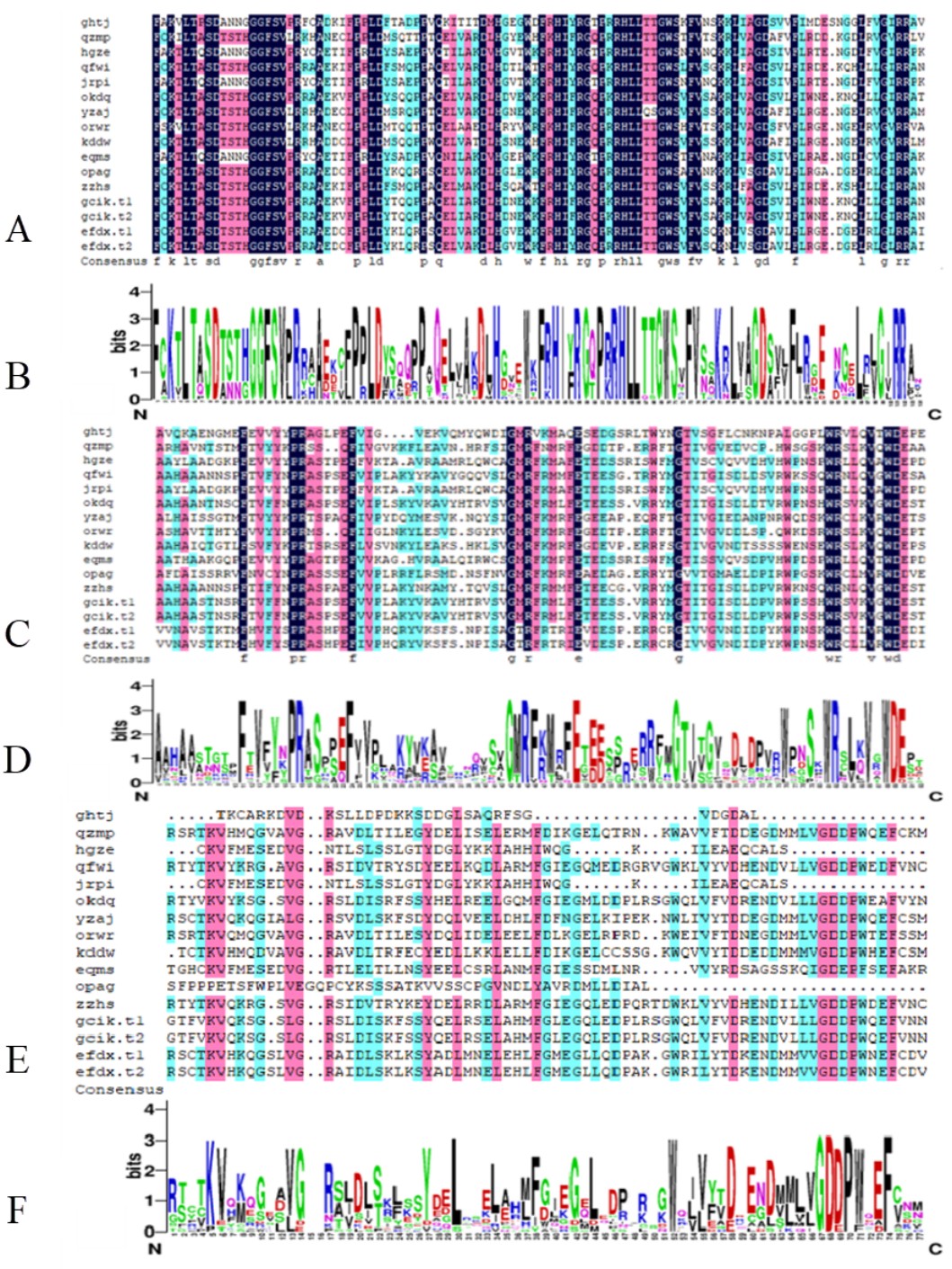

**Figure 2 Conserved domain of BvARF proteins.** (A) Alignment result of B3 domain. (B) Weblogo of B3 domain. (C) Alignment result of AUX_RESP domain. (D) Weblogo of AUX_RESP domain (E) Alignment result of AUX/IAA domain. (F) Weblogo of AUX/IAA domain.

**Table 2  Basic information of BvARF.**

| ARF name | NCBI Reference Sequence | Gene ID | Description |
| --- | --- | --- | --- |
| ghtj | XP_010672559.1 | 1108870177 | PREDICTED: auxin response factor 17 [Beta vulgaris subsp. vulgaris] |
| qzmp | XP_010696146.1 | 1108870906 | PREDICTED: auxin response factor 18 [Beta vulgaris subsp. vulgaris] |
| hgze | XP_010692716.1 | 1108784410 | PREDICTED: auxin response factor 5 [Beta vulgaris subsp. vulgaris] |
| qfwi | XP_010669598.1 | 1108883137 | PREDICTED: auxin response factor 19 [Beta vulgaris subsp. vulgaris] |
| jrpi | XP_010693396.1 | 1108789780 | PREDICTED: auxin response factor 18 isoform X2 [Beta vulgaris subsp. vulgaris] |
| okdq | XP_010675199.1 | 1108911686 | PREDICTED: auxin response factor 8 [Beta vulgaris subsp. vulgaris] |
| yzaj | XP_010678349.1 | 1108924588 | PREDICTED: auxin response factor 2 [Beta vulgaris subsp. vulgaris] |
| orwr | XP_010695583.1 | 1108811087 | PREDICTED: auxin response factor 9 [Beta vulgaris subsp. vulgaris] |
| kddw | XP_010682136.1 | 1108942544 | PREDICTED: auxin response factor 1 [Beta vulgaris subsp. vulgaris] |
| eqms | XP_010684071.1 | 1108952427 | PREDICTED: auxin response factor 18 [Beta vulgaris subsp. vulgaris] |
| opag | XP_010686391.1 | 1108960968 | PREDICTED: auxin response factor 3 isoform X2 [Beta vulgaris subsp. vulgaris] |
| zzhs | XP_010687536.1 | 1108964617 | PREDICTED: auxin response factor 7 isoform X1 [Beta vulgaris subsp. vulgaris] |
| gcik.t1 | XP_010688802.1 | 1108976183 | PREDICTED: auxin response factor 6 isoform X2 [Beta vulgaris subsp. vulgaris] |
| gcik.t2 | XP_010688801.1 | 1108976178 | PREDICTED: auxin response factor 6 isoform X1 [Beta vulgaris subsp. vulgaris] |
| efdx.t1 | XP_010689575.1 | 1108976862 | PREDICTED: auxin response factor 4 isoform X1 [Beta vulgaris subsp. vulgaris] |
| efdx.t2 | XP_010689577.1 | 1108976872 | PREDICTED: auxin response factor 4 isoform X3 [Beta vulgaris subsp. vulgaris] |

of transcript variants of BvARF gene efdx. Therefore, a total of 14 BvARF gene loci were obtained and indicated by the name in the sugar beet genome-wide database.

The 16 BvARF proteins were blastp aligned with the NCBI database and all found to be members of the BvARF family, as shown in Table 2. No more new BvARF was identified. Descriptions in NCBI suggest that jrpi, opag and zzhs may have isoforms.

## Analysis of physicochemical properties of BvARF proteins

Sequence analysis showed that the average length of the coding region of ARF gene was 2,390 bp (1,734–3,429 bp), the average number of amino acids encoding proteins was 796 (577–1,142), and the average molecular weight predicted by ExPASY was 88.7 kDa (64,060.51–127,711.63 Da), the isoelectric point averages 6.13 (5.21–7.9), as shown in Table 3.

**Table 3 Alignment result of BvARF proteins with NCBI non-redundant protein sequences database.**

| ARF name | ORF (bp) | Amino acid | Molecular weight (Da) | PI |
|---|---|---|---|---|
| ghtj | 1,734 | 577 | 64,060.51 | 6 |
| qzmp | 2,007 | 668 | 74,885.18 | 6.23 |
| hgze | 2,631 | 876 | 97,074.28 | 5.21 |
| qfwi | 3,249 | 1,082 | 119,552.61 | 6.19 |
| jrpi | 1,806 | 601 | 67,120.47 | 7.9 |
| okdq | 2,442 | 813 | 90,362.34 | 5.92 |
| yzaj | 2,523 | 840 | 94,146.56 | 6.33 |
| orwr | 1,917 | 638 | 71,317.15 | 5.47 |
| kddw | 1,980 | 659 | 73,156.89 | 5.68 |
| eqms | 2,088 | 695 | 76,649.34 | 6.28 |
| opag | 2,073 | 690 | 76,782.42 | 6.57 |
| zzhs | 3,429 | 1,142 | 127,711.63 | 6.13 |
| gcik.t1 | 2,796 | 931 | 104,483.1 | 6.03 |
| gcik.t2 | 2,805 | 934 | 104,827.53 | 6.03 |
| efdx.t1 | 2,400 | 799 | 89,067.37 | 6.04 |
| efdx.t2 | 2,367 | 788 | 88,000.25 | 6.11 |

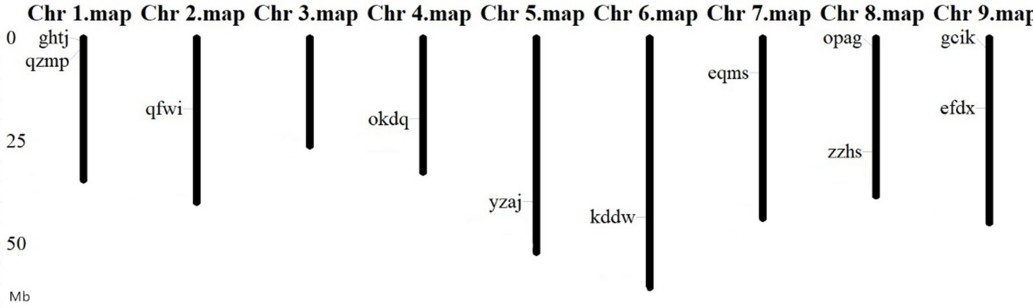

**Figure 3 Chromosomal localization of BvARF genes.** The unit of gene position is Mb Phylogenetic relationships and gene structures analysis of BvARF.

## Chromosomal localization of BvARF gene

The location analysis showed that 11 BvARF genes were distributed in 8 chromosomes except chromosome 3. There are 2 BvARF on chromosomes 1, 8, and 9, and one on chromosomes 2, 4, 5, 6, and 7, as shown in Fig. 3. The location information of other three genes hgze, jrpi, orwr is unknown in the database, that may be caused by these BvARF genes were located in the scaffolds which could not be assembled in the chromosomes.

## Phylogenetic relationships and gene structures analysis of BvARF

To survey the evolutionary relationships between BvARF proteins from sugar beet and previously-reported ARF proteins from the dicot *Arabidopsis* we constructed an unrooted phylogenetic tree with 16 BvARF and 23 *Arabidopsis* ARF protein sequences. The results are

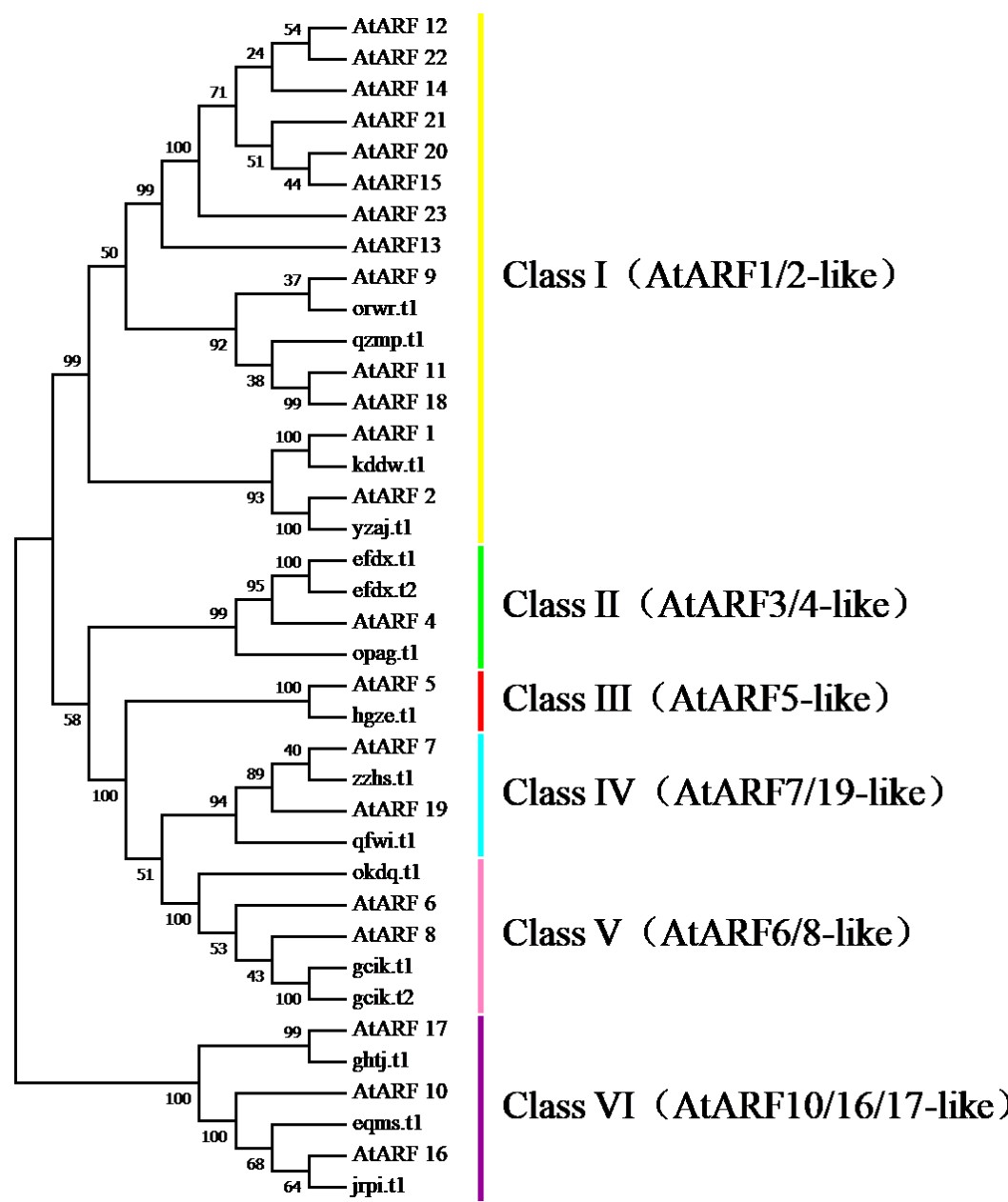

**Figure 4** Phylogenetic relationships of BvARF and AtARF proteins.

shown in Fig. 4, based on existing descriptions of ARF conserved domains in *Arabidopsis* (*Hagen & Guilfoyle, 2002*), and all 39 ARF proteins were divided into six categories.

The full-length sequence of the 16 BvARF proteins screened by MEGA7 was used to make a rootless tree by the neighbor-joining method, and the bootstrap procedure was tested with 1000 bootstrap repetitions. The results are shown in Fig. 5, which are also divided into six categories: Class I (AtARF1/2-like), Class II (AtARF3/4-like), Class III (AtARF5-like), Class IV (AtARF7/19-like), Class V (AtARF6/8-like) and Class VI (AtARF10/16/17-like).

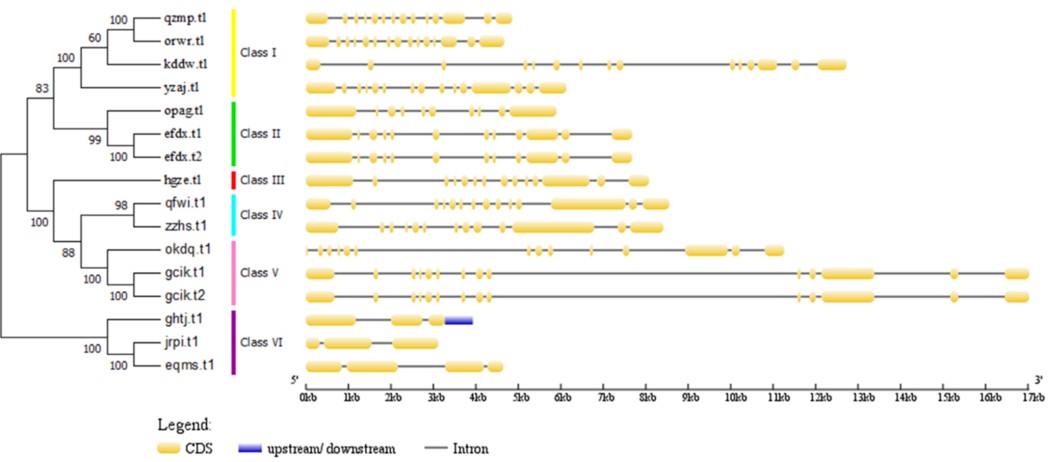

**Figure 5** **Phylogenetic relationships of BvARF proteins and exon/intron stuctures of BvARF gene.** The number on the node represents the confidence value of the branch; the gene class is represented in a different color on the right side of the rootless tree. Exon/intron stuctures of the BvARF gene. Exons and introns are represented by yellow box and black lines, respectively.

Four members were assigned to Class I (qzmp, orwr, kddw and yzaj) 2 to Class II (opag and efdx), Class III only contains hgze, 2 to Class IV (qfwi and zzhs), 2 to Class V (okdq and gcik), and 3 to Class VI (ghtj, jrpi and eqms). The intron/exon structure analysis of the BvARF genes can provide valuble imformation concerning evolutionary relationships among taxa. Analysis of these sequences revealed that all BvARF genes contained introns in the coding sequence, and the number of exons varied from 3 to 15. In general, most of the BvARF members within a given group possessed a similar exon/intron structure in terms of intron numbers and exon length.

## Motifs analysis and subcellular localization prediction of BvARF proteins

For the motifs analysis of the BvARF proteins, the expected number of motifs was set to 20. The motifs were sorted according to the e-value from small to large, and the protein sequences were ranked according to the reliability of the predicted motifs. The results are shown in Figs. 6 and 7. Some motifs were common to all BvARF proteins, such as motifs 1, 2, 3, and 5, and some motifs were specific to certain proteins, such as motifs 13, 18, and 20. Comparing the results of the motifs analysis with the conserved domains of the previous Pfam, it was found that the motif 1, 2 constituted the B3 domain, the motif 6, 8, 11, 12 constituted AUX_RESP, and the motif 4, 9, 14 constituted AUX/IAA. Previous studies have shown that ARF activates or inhibits target genes in relation to amino acids in the middle region (*Tiwari, 2003*; *Ulmasov, Hagen & Guilfoyle, 1999*), and it is worth noting that phantom 17 has a dense glutamine (Q) composition. It has shown in previous research that when glutamines gather in the middle region of the ARF proteins were generally considered to have activation domain activity (*Shen et al., 2015*). It plays a transcriptional activation role in the transcriptional regulation of auxin, and further predicts that qfwi, zzhs, okdq, gcik may have transcriptional activation which were all in Class IV and Class V.

MOTIFS

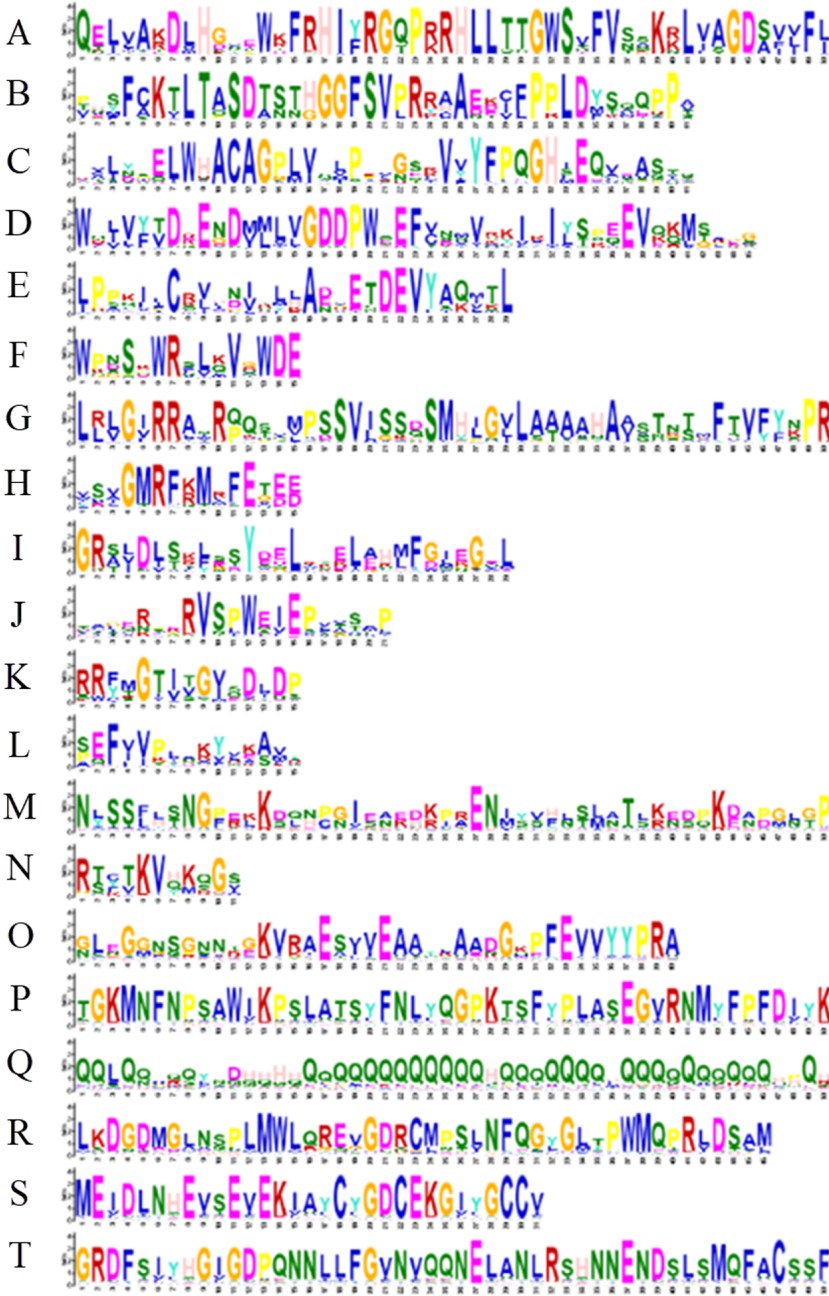

**Figure 6  Motifs in BvARF proteins.** The motifs were arranged according to the e-value from small to large, the letters in each motif were amino abbreviation. The size of the letter represented the saliency of the amino acid in the motif. The larger the letter, the higher the saliency, which is, the higher the frequency at which the amino acid appears in the same position in the same motif in different sequences.

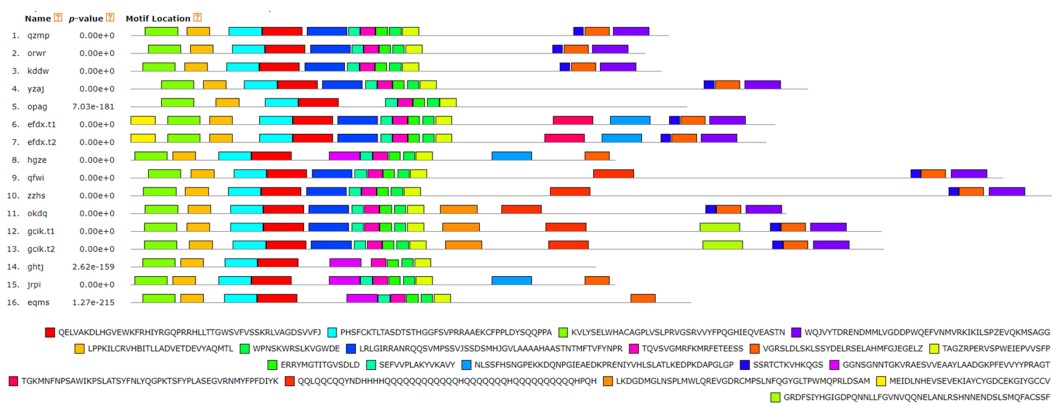

**Figure 7  Analysis of BvARF proteins motif.** The different color blocks correspond to different motifs. The width of the color block is the length of the motif. The height of the color block represents the saliency of the motifs in the sequence. The higher the saliency, the more able to match the predicted motifs.

The subcellular localization prediction of CELLO on BvARF proteins is shown in Table 4. The results showed that the most of BvARF proteins were only scored in the nucleus, that is, they were all more reliably located in the nucleus. The scores of hgze and jrpi were high in cytoplasm and nucleus. Subcellular localization results indicate that almost all BvARF proteins play a regulatory role in the nucleus.

## Gene ontology annotation

To survey the functions of the BvARF, GO annotation was obtained from PlantTFDB to construct GO graphs. As shown in Fig. 8, all BvARF (14, 100%) are involved in cell part, cell and organelle. The results also showed that the BvARF were involved in diverse biological processes and predominantly participated in metabolic process, biological regulation, regulation of biological process, cellular process(12, 85.7%), binding(11, 78.6%) and response to stimulus(8, 57.1%). The GO terminologies of BvARF were relatively concentrated.

## Expression profiles of BvARF genes in sugar beet under normal growth conditions and in response to salinity stress

In recent years, it has reported that auxin acts as integral part of plant response to salinity stress. Thus we compare the levels of expression of BvARF in condition with and without salt treatment. Expression patterns of 14 BvARF in leaves and root were analyzed by CFX96 Real-Time System to clarify the roles of BvARF genes in salt tolerance. The results are shown in Figs. 9 and 10. In the leaves, except for jrpi, yzaj and eqms, other BvARF genes all showed different degrees of up-regulation or down-regulation. Most of them showed a down-regulation, and in which the degree of 6 BvARF were significantly reduced (ghtj, okdq, opag, zzhs and efdx). And a few BvARF genes were up-regulated, with only upward trend of gcik was obvious. In the root, after treatment with salinity stress, except for qfwi, orwr and eqms, other BvARF genes all showed different degrees of up-regulation or down-regulation. Most of the BvARF genes showed an upward trend, and in which 6

**Table 4  Subcellular localization of BvARF proteins.**

| BvARF proteins | Prediction scores | | | | | | | | | | | |
|---|---|---|---|---|---|---|---|---|---|---|---|---|
| | Extracellular | Plasma membrane | Cytoplasmic | Cytoskeletal | ER | Golgi | Lysosomal | Mitochondrial | Chloroplast | Peroxisomal | Vacuole | Nuclear |
| qzmp | 0.067 | 0.073 | 0.739 | 0.011 | 0.026 | 0.06 | 0.015 | 0.124 | 0.179 | 0.032 | 0.023 | 3.652[*] |
| orwr | 0.032 | 0.027 | 0.571 | 0.007 | 0.012 | 0.049 | 0.004 | 0.088 | 0.159 | 0.018 | 0.005 | 4.028[*] |
| kddw | 0.074 | 0.044 | 0.404 | 0.014 | 0.051 | 0.075 | 0.016 | 0.158 | 0.354 | 0.039 | 0.027 | 3.744[*] |
| yzaj | 0.128 | 0.048 | 0.561 | 0.019 | 0.012 | 0.026 | 0.008 | 0.145 | 0.056 | 0.044 | 0.009 | 3.945[*] |
| opag | 0.094 | 0.076 | 1.135 | 0.064 | 0.024 | 0.024 | 0.044 | 0.164 | 0.507 | 0.112 | 0.03 | 2.726[*] |
| efdx.t1 | 0.15 | 0.073 | 0.995 | 0.022 | 0.03 | 0.038 | 0.042 | 0.232 | 0.281 | 0.091 | 0.022 | 3.024[*] |
| efdx.t2 | 0.148 | 0.078 | 0.906 | 0.024 | 0.029 | 0.038 | 0.046 | 0.302 | 0.313 | 0.071 | 0.022 | 3.022[*] |
| hgze | 0.143 | 0.248 | 1.590[*] | 0.016 | 0.033 | 0.092 | 0.101 | 0.54 | 0.842 | 0.179 | 0.064 | 1.153[*] |
| qfwi | 0.106 | 0.369 | 0.2 | 0.03 | 0.016 | 0.009 | 0.008 | 0.027 | 0.012 | 0.008 | 0.009 | 4.207[*] |
| zzhs | 0.219 | 0.811 | 0.304 | 0.046 | 0.052 | 0.014 | 0.016 | 0.059 | 0.032 | 0.024 | 0.015 | 3.409[*] |
| okdq | 0.285 | 0.347 | 0.611 | 0.016 | 0.027 | 0.021 | 0.087 | 0.07 | 0.042 | 0.044 | 0.019 | 3.430[*] |
| gcik.t1 | 0.175 | 0.388 | 0.423 | 0.021 | 0.036 | 0.018 | 0.029 | 0.054 | 0.021 | 0.02 | 0.011 | 3.803[*] |
| gcik.t2 | 0.197 | 0.401 | 0.411 | 0.02 | 0.035 | 0.019 | 0.032 | 0.051 | 0.021 | 0.02 | 0.011 | 3.782[*] |
| ghtj | 0.131 | 0.122 | 1.376 | 0.016 | 0.02 | 0.032 | 0.074 | 0.212 | 0.64 | 0.129 | 0.064 | 2.184[*] |
| jrpi | 0.143 | 0.248 | 1.590[*] | 0.016 | 0.033 | 0.092 | 0.101 | 0.54 | 0.842 | 0.179 | 0.064 | 1.153[*] |
| eqms | 0.17 | 0.117 | 1.049 | 0.011 | 0.041 | 0.032 | 0.047 | 0.163 | 0.447 | 0.042 | 0.021 | 2.862[*] |

**Notes.**

[*]Confidence score is significantly higher than other, which means higher credibility.

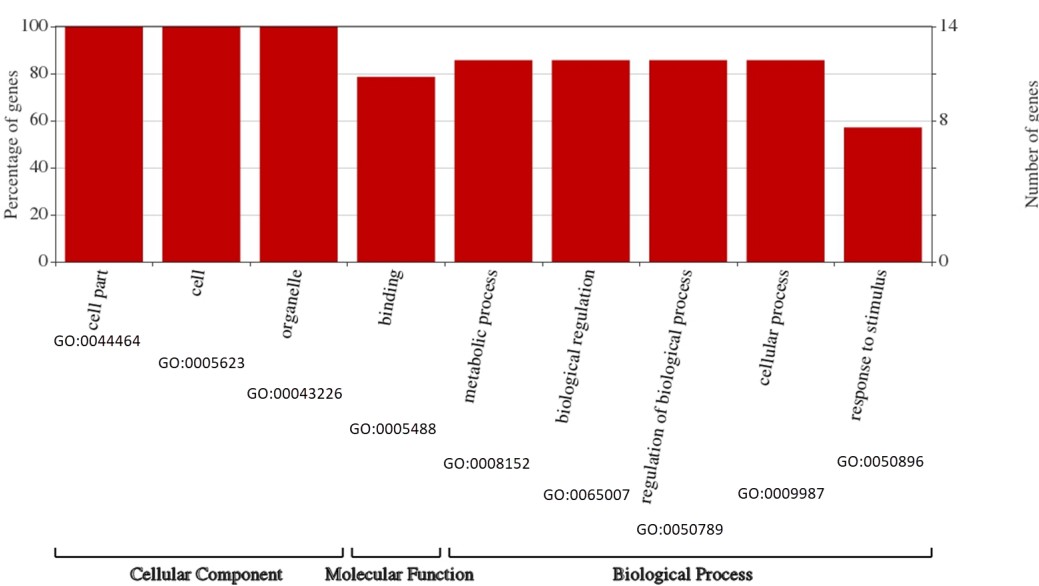

**Figure 8** **Assignment of GO categories to BvARF genes.** Expression profiles of BvARF genes in sugar beet under normal growth conditions and in response to salinity stress.

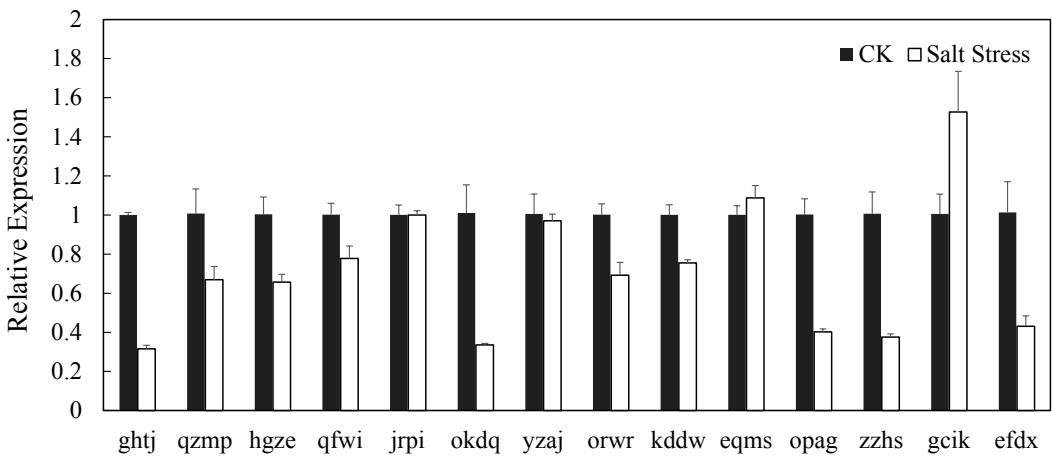

**Figure 9** **Expression analysis of BARF genes in leaf in response to salinity stress.**

BvARF showed an obvious upward trend (hgze, okdq, yzaj, kddw, gcik and efdx). And 5 BvARF genes showed a downward trend, but the degree of down-regulation was not significant (ghtj, qzmp, qfwi, jrpi and eqms).

## DISCUSSION

Sugar beet is mainly distributed in Northwest, Northeast and North China, where the saline-alkali soil area is large. Therefore, it is particularly important to study the salt tolerance mechanism of sugar beet and cultivate new varieties. Members of the ARF family

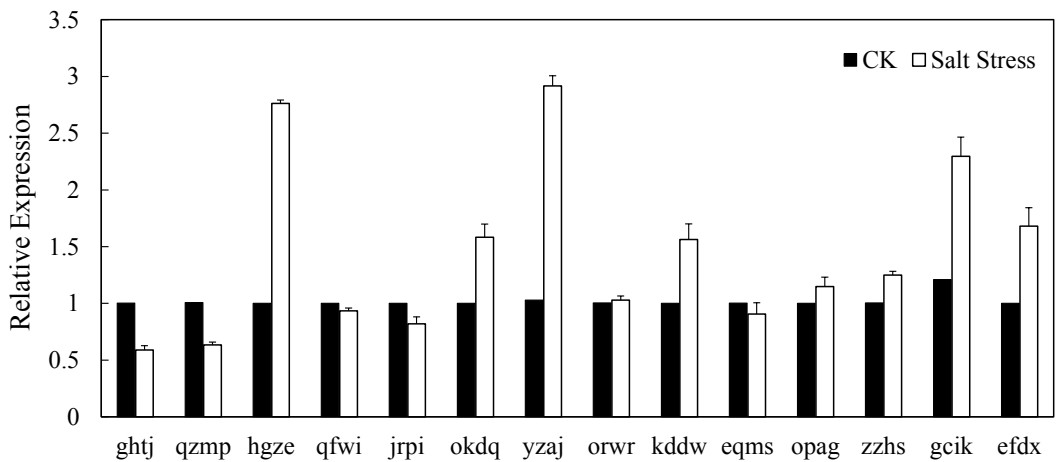

**Figure 10** **Expression analysis of BvARF genes in root in response to salinity stress.**

have been reported to be involved in regulation of plant growth, development and response to abiotic stresses. However, the ARF family members in sugar beet have not been studied previously. Thus in order to shed light on the roles of BvARF genes in response salt stress, we delineated the major structural characteristics and expression profiles of ARF genes in this species.

In this study, 16 BvARF proteins were discovered by genome-wide analysis, which were transcribed by 14 BvARF genes. The number of members of the BvARF gene family was far less than which of *Arabidopsis* (23), maize (31), rice (25) and other plants. The lack of clustering in spatial locations indicates that there is less gene duplication in the BvARF gene in the long-term evolution process (*Cannon et al., 2004*; *Tang et al., 2018*).

The phylogenetic comparison of ARF proteins has been widely used in many species, and these proteins have been widely reported in different species and evolutionary relationships. In , according to the phylogenetic tree analysis, BvARF proteins and AtARF were divided into six categories. Four BvARF proteins with similar gene structures belong to the Class I ARF, whereas 13 AtARFs were assigned to this group. This suggests that the genes of the group may have been either lost from the sugar beet lineage or acquired in the *Arabidopsis* after divergence from the last common ancestor shared by Arabidopsis and sugar beet. And *Arabidopsis* I ARF-like is closely related to the development of flower and seedling, suggesting that these four ARF may have similar functions in *beta vulgaris* (*Ellis et al., 2005*). Three BvARF proteins belong to Class II ARF, and according to the Class II ARF function in *Arabidopsis*, which can affect the dorsal ventrality of lateral organs (*Pekker, Alvarez & Eshed, 2005*). One BvARF protein belongs to Class III ARF, suggesting that it may play a regulatory role in stomatal development and petal differentiation(*Cole et al., 2009*). Two BvARF proteins belong to Class IV ARF and may regulate lateral roots (*Okushima et al., 2005*). Three BvARF proteins belong to Class V and may be involved in the regulation of auxin homeostasis (*Tian et al., 2004*). Three BvARF proteins belonging to Class VI may regulate the formation of pollen walls (*Wang et al., 2017*).

The localization of all 16 BvARF proteins was predicted in the nucleus, indicating their roles in transcriptional regulation. All 16 BvARF proteins have both a B3 DNA binding domain and an ARF response domain, moreover 5 BvARF proteins lack of a CTD which allows dimerization of the ARF proteins and the Aux/IAA proteins. The percentage of members missing CTD (31.3%) was similar to other species, such as tea trees (26.7%) (*Xu et al., 2016*) and tomatoes (28.6%) (*Kumar, Tyagi & Sharma, 2011*).

The exon intron pattern shows the homology between species to some extent. The exon-intron splicing arrangement and intron numbers in the BvARF genes in the sugar beet genome were similar to other plants. It is worth noting that BvARF genes in Class VI is consistent with the AtARF genes in Class VI, and the exons are long and are less separated by introns. Motif analysis indicated that the motif 1, 2, 4, 6, 8, 9, 11, 12 and 14 present in most of the BvARF proteins, are the typical motifs present in the auxin response factors protein. Motif 1 and 2 are involved in DNA binding, while 4, 9 and 14 are involved in protein-protein interaction. ARF in a group often has a common motif belonging to their own group. Generally, most of the BvARF genes in one group had similar gene structure and conserved motifs, which further supports their classification as described here and the evolutionary relationships among the groups. Previous studies have shown that the regulation of ARF activation or inhibition is mainly determined by the type of amino acids in the middle region. Belong to Class IV and V, four BvARF (qfwi, zzhs, okdq, gcik) have a poly-Q motif in the central region, indicating that they have an activation effect.

ARF is a kind of transcription factor that regulates the expression of auxin-responsive genes. The expression of ARF is also affected by biological and abiotic stresses (*Hannah, Heyer & Hincha, 2005*; *Jain & Khurana, 2009*; *Kang et al., 2018*; *Yu et al., 2017*). Quantitative results of BvARF genes under salt stress allowed us to identify BvARF genes involved in stress response. The results suggested that six of the 14 BvARF genes were induced or inhibited by salt in leaves and six were induced in roots. And some BvARF expressed differently in leaves and roots, such as okdq and efdx, while some showed the same trend, such as gcik. It can be speculated that ARF have played different degrees of regulation on the salt tolerance of sugar beet. Our study provides evidence that BvARF may participate in sugar beet response to salt stress. Further research is needed to determine which mechanism to achieve salt tolerance.

## CONCLUSION

In this study, a total of 16 BvARF protein were identified, and we analyzed the gene structures, chromosome location, protein conserved domains, phylogeny, proteins motif, subcellular localization and gene ontology consortium etc. by bioinformatics method. And quantitatively comparing the expression of BvARF in leaves and roots under salt stress and normal environment. This study lays a foundation for further study of the functional characteristics of BvARF, and provided a new idea for breeding salt-tolerant sugar beet varieties.

### Funding

This work was supported by the National Natural Science Foundation of China (No. 31571731), and the Sugar Industry Modern Industrial Technology System Construction Project of Ministry of Agriculture (NO. CARS-170111-06 and NO. CARS-170104-01). The funders had no role in study design, data collection and analysis, decision to publish, or preparation of the manuscript.

### Grant Disclosures

The following grant information was disclosed by the authors:
National Natural Science Foundation of China: 31571731.
Sugar Industry Modern Industrial Technology System Construction Project of Ministry of Agriculture: CARS-170111-06, CARS-170104-01.

### Competing Interests

The authors declare there are no competing interests.

### Author Contributions

- Jie Cui conceived and designed the experiments, authored or reviewed drafts of the paper, and approved the final draft.
- Xinyan Li performed the experiments, analyzed the data, prepared figures and/or tables, and approved the final draft.
- Junliang Li and Congyu Wang analyzed the data, authored or reviewed drafts of the paper, and approved the final draft.
- Dayou Cheng performed the experiments, prepared figures and/or tables, and approved the final draft.
- Cuihong Dai conceived and designed the experiments, prepared figures and/or tables, technical support, and approved the final draft.

### Data Availability

The raw measurements are available as a Supplementary File.

## Supplemental Information

Supplemental information for this article can be found online at http://dx.doi.org/10.7717/peerj.9131#supplemental-information.

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
