# Peer review of "Genome-wide sequence identification and expression analysis of ARF family in sugar beet (Beta vulgaris L.) under salinity stresses"

_PeerJ, doi:10.7717/peerj.9131_

## Round 0.1 · original submission · Major Revisions

Dear Xinyan,

I now have two review report back and would like to share with you. The reviewers appreciate your interesting work. But there are a number of issues needed to be addressed before we can accept for publication in the journal. I would suggest that you and your colleagues read through these comments and make changes on your manuscript as suggested.

I look forward to your updated version.

Kind regards,

Zemin

Reviewer 1 ·

Basic reporting

No comment.

Experimental design

No comment.

Validity of the findings

No comment.

Additional comments

In this work, the authors identified 16 auxin response factor related genes using sugar beet genome sequence and gene annotations. Bioinformatics analyses, e.g., exon-intron structure, chromosomal locations, expression profiling, were conducted. Some BvARF genes in sugar beet were found to be up-regulated or down-regulated to varying degrees under salinity stress.
Generally, the work is lack of novelty. Some results can be searched by publically database of the genome and most of the manuscript is descriptive. More functional evidences of these genes need to be provided. Some suggestions:
1. There were no details for the salinity stress experiments, which should be added in the Method section.
2. Several control genes should be added. According to Fig. 9 and Fig. 10, I don't think the expressions showed large changes.
When genes were picked from another gene family, probably another gene set will have more remarkable response to salinity stress.
3. Any transgenic experiments of the BvARF into Arabidopsis to check the phenotypes with salinity stress?
4. The genes (ghtj, qzmp, hgze ...) seem to be named at will.

Reviewer 2 ·

Basic reporting

The authors of this manuscript describe the identification and analyses on the auxin response factor (BvARF) gene family in the sugar beet genome using a bioinformatics approach. Using a quantitative real-time PCR approach, they have also shown the changed levels of gene expression in many of these BvARFs when plant seedlings have been under salinity treatment. I believe this work would add to the understanding of the plant auxin response factors.

This manuscript is well organized. The figures and tables are clear and helpful.

In general, the English language should be improved to ensure that international readers can clearly understand the text. Some examples are given below in more detail.

The introduction requires more detail and accuracy.
• For example, I suggest that the authors improve the description at lines 33- 36 to provide more details about what has been known on the ARFs of other plants and justification for your study.
• At line 37, I suggest the authors check if the statement “Sugar beet is a biennial herb of Chenopodium” is true. I am not an expert in plant systemics, but I would think sugar beet belongs to a different genus.
• At lines 39-40, more details required and reference needed for “In China, the sugar beet production area and the saline land overlap in a large area.” as this seems to be the only justification of the saline tolerance study.

Experimental design

The experimental design was clear enough. I can’t judge if the saline tolerance study design was adequate since it is not my area of expertise. But the treatment seems rather simplistic.

Some changes, additions, modifications would be suggested as follows:
1) lines 53-55: It is not clear what the germination conditions (the settings for the incubator and culturing conditions). This should be clarified.
2) lines 61-62: it is not clear which dataset(s), ie annotation or genomic, and which version were downloaded and searched. Please clarify.
3) line 94: Please define ICDH.
4) Line 103: It is considered untidy to start sentences with figures. Please either reword your sentence or write the number in full.
5) Line 117: Should it be “…a total of 14 BvARF gene loci were obtained….”
6) lines 126-127: Surely all genes can be located to a genome assembly, either in chromosomes or scaffolds. So it is incorrect to say “…The other three genes hgze, jrpi, orwr could not be located, …” Please clarify/
7) lines 133-134: please specify “other plants” and provide background in the Introduction. The criteria for dividing into six categories need to be clarified.
8) lines 149-151: what program was used for the motif analysis? Please justify why the expected number of motifs was set to 20.

Validity of the findings

The results are reasonable given the experiments. However, I suggest the authors moderate their claims in lines 246-248 that “…BvARF may play an important role in sugar beet response to salt stress…” purely based on different expression levels. A lot more supporting evidence is required for that claim.

Additional comments

It is a paper that would add more knowledge to the understanding of plant auxin response factors, even though similar studies have been published in other plant species.

Changes both with the English language and the text will be required to improve the current manuscript to the publication standards. Some examples have been given below.

The introduction requires more detail and accuracy.
• For example, I suggest that the authors improve the description at lines 33- 36 to provide more details about what has been known on the ARFs of other plants and justification for your study.
• At line 37, I suggest the authors check if the statement “Sugar beet is a biennial herb of Chenopodium” is true. I am not an expert in plant systemics, but I would think sugar beet belongs to a different genus.
• At lines 39-40, more details required and reference needed for “In China, the sugar beet production area and the saline land overlap in a large area.” as this seems to be the only justification of the saline tolerance study.

Some changes, additions, modifications would be suggested:
• lines 53-55: It is not clear what the germination conditions (the settings for the incubator and culturing conditions). This should be clarified.
• lines 61-62: it is not clear which dataset(s), ie annotation or genomic, and which version were downloaded and searched. Please clarify.
• line 94: Please define ICDH.
• Line 103: It is considered untidy to start sentences with figures. Please either reword your sentence or write the number in full.
• Line 117: Should it be “…a total of 14 BvARF gene loci were obtained….”
• lines 126-127: Surely all genes can be located to a genome assembly, either in chromosomes or scaffolds. So it is incorrect to say “…The other three genes hgze, jrpi, orwr could not be located, …” Please clarify/
• lines 133-134: please specify “other plants” and provide background in the Introduction. The criteria for dividing into six categories need to be clarified.
• lines 149-151: what program was used for the motif analysis? Please justify why the expected number of motifs was set to 20.

I also suggest the authors moderate their claims in
lines 246-248 that “…BvARF may play an important role in sugar beet response to salt stress…” purely based on different expression levels. A lot more supporting evidence is required for that claim.

---

## Round 0.2 · Minor Revisions

Dear Dr Li,

One of our Section Editors checked your manuscript and made this suggestion:

“As the sugar beet genome already has functional annotation, were any new ARF genes identified here? Please provide a table relating the annotation revealed by the HMMR analysis to the existing annotation.”

I would be grateful if you could follow the suggestion and provide the table.

Thank you again and look forward to your updated version soon.

Zemin

Reviewer 1 ·

Basic reporting

No comment.

Experimental design

No comment.

Validity of the findings

No comment.

Additional comments

The manuscript has been revised according to my suggestions. I'm satisfied with the revision.

Reviewer 2 ·

Basic reporting

The revised manuscript has improved overall and satisfied the required standards for publication in its current form.

Experimental design

All corrections are satisfactory in the revised manuscript.

Validity of the findings

The change made by the authors in the revised manuscript is satisfactory.

Additional comments

The changes made by the authors in the revised manuscript has improved and strengthened their research work to the publication standard in this journal and would add more knowledge to the understanding of plant auxin response factors. I recommend it to be accepted for publication.

---

## Round 0.3 · Minor Revisions

Dear Dr. Cui and Dr. Li,

Thanks for the new version. Our Section Editor has looked at the manuscript and made these comments:

"“The authors need to make the data that their describing more accessible to the readers. I see figures with pictures of partial sequence data, but what is needed is a pointer to the reference sequence with coordinates from a readily available resource, or to provide as supplemental data the sequence data that was used in this study for comparison. Readers sometimes have resources available which can best validate the work by others. And, once again, since they are try to add new annotation to data being studied, a community annotation would be expected. There is mention of GO terms, but the actual GO numbers defining their extracted properties is not included. This would be included within a table of some sort. I don’t agree at this stage. I would recommend modification to help give the readers something to work with if they are to develop this story line for this gene family.”

Please follow his suggestions and make changes before submitting a new version.

Many thanks for all your great efforts!

Zemin

---

## Round 0.4 · Minor Revisions

Dear Dr. Cui and Dr. Li,

Thank you both again for the updated version. There are some comments from, Gerard Lazo, the Section Editor, who suggested some small changes:

“The set of genes studied is very limited to the 16 followed and perhaps would have been served better in a larger panel using a microarray-type format for validation.

However, with the defined set and additional suggestions followed, the manuscript appears acceptable for publication; however, there may be further points to bring clarity to the manuscript. Though a single germplasm selection was used for testing, it would be curious to note the contrasting signaling which is lacking in the experimental design
(perhaps an error to reject the manuscript) as no history is presented in the background information for the O68 germplasm. A better description of whether the germplasm used is a salt-tolerant line or not should be further explained. I will suggest a minor modification before clearing the manuscript as accepted.

There were also a few lines that may need attentions (listed below):

EDITS LINE NO: / BEFORE / AFTER / [COMMENTS]

LINE 12/13: / remarkably response / . / [domain?]

LINE 31: / mediated / . / [?]

LINE 38: / researches / . / [?]

LINE 50: / literatures / . / [?]

LINE 75: / genome-wild / genome-wide / [.]

LINE 196: / It have shown / . / [?]

These are basically minor edits required before moving forward.”

I would be grateful if you could make these changes?

Kind regards,

Zemin

---

## Round 0.5 · accepted · Accept

Many thanks and happy to accept for publication.

Reviewer 1 ·

Basic reporting

No comment.

Experimental design

No comment.

Validity of the findings

No comment.

Additional comments

I'm satisfied with the revision now.

Reviewer 2 ·

Basic reporting

The inclusion of GO terms and sequences in the supplemental files has further improved the accessibility of the results and sequence data used. It has addressed editor's concerns and satisfied the required standards for publication.

Experimental design

Satisfactory in the revised manuscript.

Validity of the findings

Satisfactory in the revised manuscript.

Additional comments

The inclusion of GO terms and sequences in the supplemental files has further improved the accessibility of the results and sequence data. It has addressed editor's concerns and satisfied the required standards for publication. The revised manuscript has further improved and strengthened their research work to the publication standard in this journal and would add more knowledge to the understanding of plant auxin response factors. I recommend it to be accepted for publication.